# The Relationship between Nutrient Intake and Cataracts in the Older Adult Population of Korea

**DOI:** 10.3390/nu14234962

**Published:** 2022-11-23

**Authors:** Sangyun Lee, Soyeon Lee, Myeonghyeon Jeong, Sunwoo Jung, Myoungjin Lee, Sunyong Yoo

**Affiliations:** 1Department of IoT Artificial Intelligence Convergence Major, Chonnam National University, Gwangju 61186, Republic of Korea; 2Department of ICT Convergence System Engineering, Chonnam National University, Gwangju 61186, Republic of Korea

**Keywords:** nutrition surveys, nutrients, lens, cataracts in the elderly, cataract risk factors, Korean National Health and Nutrition Examination Survey

## Abstract

Cataracts are a prevalent ophthalmic disease worldwide, and research on the risk factors for cataracts occurrence is actively being conducted. This study aimed to investigate the relationship between nutrient intake and cataracts in the older adult population in Korea. We analyzed data from Korean adults over the age of 60 years (cataract: 2137, non-cataract: 3497) using the Korean National Health and Nutrition Examination Survey. We performed univariate simple and multiple logistic regressions, adjusting for socio-demographic, medical history, and lifestyle, to identify the associations between nutrient intake and cataracts. A higher intake of vitamin B1 in the male group was associated with a lower incidence of cataracts. A lower intake of polyunsaturated fatty acids and vitamin A, and a higher intake of vitamin B2 in the female group were associated with a higher incidence of cataracts. Our study demonstrated that polyunsaturated fatty acids, vitamin A, and vitamin B2 could affect the incidence of cataracts according to sex. The findings could be used to control nutrient intake for cataract prevention.

## 1. Introduction

Cataracts is the most common cause of reversible vision loss worldwide [1]. In 2020, cataracts represented approximately 45% of the 33.6 million cases of blindness, and over 15 million of those were diagnosed in adults aged 50 years and older [2]. In detail, cataract affects more than 22 million people and accounts for USD 6.8B in annual medical costs in the United States [3]. More than 22% of people aged 45–89 years are affected by cataracts in China [4]. In addition, cataracts is the leading cause of early visual impairment and the third leading cause of blindness in Europe [5]. It has also been reported that prevalence of cataracts in the older age group (>60 years) is 53%–58% [6]. As the worldwide population ages, the occurrence of cataracts and the number of people affected will continue to increase. According to the National Health Insurance Statistical Yearbook provided by the Korea National Health Insurance Service, 1,107,823 Koreans were treated for cataracts at hospitals in 2020. Among them, 740,899 people, about 66.9% of the total, were over the age of 65 years [7]. In addition, the number of cataract operations was found to be significantly higher than that of other major surgeries. In Korea, the number of cataract surgeries per 100,000 people increased at an average annual rate of 7.6% between 2016 and 2020 [8]. Interest in cataract prevention is growing as the prevalence of cataracts continues to increase yearly.

Several factors, such as age, the presence of diabetes, oxidation, and nutrition influence the occurrence of cataracts [9,10,11]. Among them, diet and nutrient intake have recently gained attention in preventing cataracts, as they can be controlled steadily at the individual level. For example, a high intake of vegetables, fruit, starchy foods (potatoes), fish, pulses, and nuts may reduce the risk of cataracts [12]. The antioxidant action of vitamins, abundant in fruits and vegetables, has been inversely associated with cataract risk [12]. In contrast, a high intake of meat or salt can increase the risk of cataracts [13]. Further, researchers have explored the association between nutrient intake and cataracts in several studies. The research was initially focused on antioxidants to understand how nutrients may impact cataracts, and later studies expanded to consider macronutrients such as carbohydrates [11,14,15]. Cross-sectional case-control studies demonstrated that high sodium intake might affect the occurrence of cataracts [16]. A Na+/K+ electrolyte imbalance in the aqueous humor is one of the biological mechanisms of cataracts [17]. High levels of extracellular sodium cause an influx of Na+ in the lens, attracting water ions. Therefore, it is difficult to maintain the low levels of sodium in cells required for lens transparency [18]. Others have examined whether the dietary intake of vitamins is related to cataract occurrence [19,20,21]. Overall, most previous studies evaluated the risk factors for only specific nutrients. Further, nearly all the studies were conducted using data from ten years ago and do not reflect the recent changes in dietary patterns [10,11,14,15].

This study analyzes associations between nutrient intake and cataracts among the older adult population of Korea. Data were obtained from the Korean National Health and Nutrition Examination Survey 2016–2018 (KNHANES VII), a nationally representative survey of the nutrient intake of people in the country. Cataract patients aged 60 years or older were selected from KNHANES data, and the nutrient intake information was collected using their health interview responses. Various statistical analyses were performed on a large cohort of older Korean adults to identify associations between nutrient intake and cataract.

## 2. Materials and Methods

### 2.1. Study Population

The KNHANES is a national monitoring system that investigates the health and nutritional status of South Koreans. It was implemented by the Korea Centers for Disease Control and Prevention based on the National Health Promotion Act in 1998 [22]. This nationally representative cross-sectional survey collects information on socioeconomic status, health-related behaviors, quality of life, healthcare utilization, anthropometric measures, biochemical and clinical profiles for non-communicable diseases, and dietary intakes via three surveys: a health interview, a health examination, and a nutrition survey [22,23,24]. We used data from the recently published KNHANES VII (2016–2018), which were sampled by a multi-stage clustered probability design that reflected three pieces of information: weight, layer, and cluster. We set 60 years and over as the cutoff age to analyze the relationship between nutrient intake and cataract occurrence in older adults. In the KNHANES dataset, the prevalence of cataracts in people under the age of 60 was only 2.48% (cataract: 244, total: 9803). Among them, the 50–59-year-olds had the highest prevalence, which was 6.67% (cataract: 190, total: 2848). This prevalence is significantly lower than the group of 60–69-year-olds, which had a cataract prevalence of 22.1% (cataract: 609, total: 2756) and the group of 70–79-year-olds, which had a cataract prevalence of 53.0% (cataract: 1528, total: 2881). Therefore, samples under the age of 60 were excluded due to the lack of positive cases for the statistical analysis. Due to the high prevalence of cataracts in the older population, many previous studies have also conducted analyses on those aged 60 or older [25,26,27]. Among the 24,269 individuals, we excluded those who with nonpositive weight values (*n* = 4089). Nonpositive weights indicate the number of people who did not respond to the food intake survey. Additionally, those under the age of 60 years (*n* = 9803) and those with missing values among the variables (*n* = 4743) were excluded. Finally, 5634 older adults—among whom 2137 had cataracts, and 3497 did not—were selected for the analysis. The analysis target selection procedure of this study is illustrated in Figure 1.

### 2.2. Study Variables

In this study, information on whether a cataract was present or not was used as a dependent variable. The health interview survey of KNHANES contains four questions related to a cataract diagnosis, which are closed-ended questions. Three of the questions require an answer of yes or no: (i) whether they have an ophthalmologist’s diagnosis, (ii) whether they have cataracts, (iii) whether they have a current treatment. The last question regards the age when the cataract was first detected, so participants answered with an age in number form. Among the participants, we selected those who agreed to the following questions: was the diagnosis made by an ophthalmologist, and whether cataracts are present?

We used the information on nutrient intake as independent variables. The KNHANES contains a food intake survey conducted using the 24 h dietary recall method [28], which is commonly used in nutrition research using population studies [28,29,30,31,32]. The process of recalculating the 24 h recall as nutrient intake was as follows: First, the KNHANES investigated all the food and beverages participants had consumed the day before. Survey items include food name, intake volume (mL), intake weight (g), and food condition. When processed food was included, the product name and manufacturer name of the food were included. Next, the KNHANES calculated the nutrients in each food obtained from the Korean food composition table provided by the Korea Rural Development Administration [33]. Finally, the total intake of each nutrient was calculated by adding up all the nutrients contained in the foods consumed during the day. We extracted all available nutrient variables provided by the KNHANES. The variables included the intake of water, protein, fat, saturated fatty acids, monounsaturated fatty acids, polyunsaturated fatty acids, omega-3 fatty acid, omega-6 fatty acid, cholesterol, carbohydrates, dietary fiber, sugars, calcium, phosphorus, iron, sodium, potassium, vitamin A, carotene, vitamin B1, vitamin B2, vitamin B3, vitamin B9, and vitamin C. These variables were divided into quartile groups according to the amount of intake in the regression analysis. The first quartile was classified as low intake, the second and third quartiles as middle intake, and the fourth quartile as high intake.

Information on socio-demographics, medical history, and lifestyle was used as adjustment variables. Socio-demographic variables included age (60 s, 70 s), marital status (single or married), education (divided based on high school graduation), and national basic livelihood (beneficiaries or non-beneficiaries). Medical history variables included cataract-related diseases, such as asthma, allergic rhinitis, hyperlipidemia, heart failure, and sinusitis [34,35,36,37,38]. Lifestyle variables included physical activity, smoking, and drinking, which were demonstrated to be associated with cataracts [39,40,41]. We collected this information from health questions from the KNHANES. Physical activity was divided into three categories: 2 h 30 min or more of moderate-intensity, 1 h and 15 min or more of high-intensity, or a mixture of moderate-intensity and high-intensity activities. A non-smoker was defined as a person who has smoked less than five packs in their lifetime or smoked in the past but does not currently smoke. A smoker was defined as a person who has smoked more than five packs in their lifetime or currently or occasionally smokes. Heavy drinking was defined as drinking more than seven cups for males or five cups for females more than twice a week.

### 2.3. Statistical Analysis

In this study, the Student’s *t*-test and Cohen’s *d* were used to analyze the basic characteristics of the study population. Both tests were conducted with the data assuming a normal distribution [42,43]. We performed a Kolmogorov–Smirnov test to verify that our data followed a normal distribution. The null hypothesis was “The distribution of our data follows a normal distribution”. The alternative hypothesis was “The distribution of our data does not follow a normal distribution”. From the result, we found that the *p*-values of all variables were greater than 0.05 (Appendix A). Therefore, the null hypothesis was not rejected. According to the null hypothesis, our data followed a normal distribution.

Through the *t*-test, it was confirmed whether there was a statistically significant difference in the mean distribution of nutrient variables between the cataract and non-cataract groups [44]. However, the *t*-test’s *p*-values were very sensitive to the sample size [44,45]. In general, it was found that the larger the sample size, the smaller the *p*-value. Therefore, Cohen’s *d*, a type of effect size, was calculated to estimate the strength of the associations through the calculation of the standardized mean difference between independent variables and a dependent variable [46]. The relationship between independent variables and a dependent variable was identified through univariate simple and multiple logistic regressions [47,48,49]. A univariate simple logistic regression evaluates the relationship between each independent variable and the one dependent variable, and a univariate multiple logistic regression evaluates the interrelationship between two or more independent variables and one dependent variable. In both simple and multiple regression, including all independent variables further helps to explore their relationship with their dependent variable. However, the use of variables that are not related to the dependent variable can be a major cause of performance degradation [50,51]. To eliminate the risk of performance degradation, only significant variables (*p* < 0.05) in the univariate simple logistic regression were used in the univariate multiple logistic regression analysis. Additionally, potential confounding variables were adjusted. Socio-demographic variables are closely related to the change in overall nutrient variables [52]. The presence or absence of medical history information may affect the prevalence of cataracts [34,35,36,37,38,53]. Furthermore, lifestyle habits such as smoking, heavy drinking, and physical activity have been demonstrated to be associated with cataract occurrence [39,40,41]. These can confound the results regardless of specific nutrient variables. To address this issue, the multiple logistic regression was performed after adjustment for covariates, such as socio-demographics, medical history, and lifestyle [54].

The multicollinearity of the independent variables was checked before the multiple logistic regression was performed. Multicollinearity occurs due to strong correlations among independent variables in a regression analysis. In this study, the multicollinearity problem was detected using the tolerance and variance inflation factor (VIF) [55,56,57]. The tolerance was defined as 1 − *R_i_*^2^, where *R_i_*^2^ was the coefficient of determination for the regression of the *i_th_* variable from the other independent variables. The VIF*_i_* was defined as the reciprocal of tolerance. Generally, *R_i_*^2^ > 0.9 can be interpreted as even if the *i_th_* independent variable is omitted, the remaining variables explain more than 90% of the dependent variable. In other words, if the VIF*_i_* value exceeds 10, the *i_th_* independent variable has a high multicollinearity problem. Both simple and multiple logistic regression analyses were performed separately according to sex. Next, the odds ratio (OR) was used to compare the relative odds of the occurrence of the outcome of interest, given the exposure to the variable of interest. The OR was defined as the following Equation (1).
(1)$$\mathrm{OR}=\frac{\mathrm{Odds}\;\mathrm{of}\;\mathrm{cataract}\;\left(\mathrm{case}\right)}{\mathrm{Odds}\;\mathrm{of}\;\mathrm{cataract}\;\left(\mathrm{reference}\right)}=\frac{\frac{P\left(\mathrm{case}\right)}{1-P\left(\mathrm{case}\right)}}{\frac{P\left(\mathrm{reference}\right)}{1-P\left(\mathrm{reference}\right)}}\;\;$$

In this equation, *p*(case) is the probability of cataract occurrence in the high- or low-level nutrient intake group and *p*(reference) is the probability of cataract occurrence in the mid-level nutrient intake group. Additionally, we calculated the *p*-values of the ORs to confirm the statistical significance of the variable for cataract occurrence. The null hypothesis was that a high- or low-level of nutrient intake had the same effect as the mid-level nutrient intake in cataract occurrence. Alternatively, the confidence interval (CI) was used to estimate the significance of the OR; if the CI excluded 1, the OR was significant. A factor analysis was used to investigate concepts that could not be easily and directly measured through the available nutrient variables. The factor analysis was used to reduce a large number of variables into fewer factors by summarizing the common variations of the variables. In general, determining the number of factors was based on the eigenvalue. A factor with an eigenvalue larger than one denoted that the variance between the variables was much more than the difference between a single variable [58]. We drew a scree plot based on the eigenvalues of all the factors to find the number of factors that had eigenvalues larger than one. The factors obtained in the initial extraction phase were often difficult to interpret because of significant cross-loadings, in which many factors correlated with many variables. Factor rotation minimized the complexity of the factor loadings and made the structure simpler to interpret. In this study, we used orthogonal Varimax rotation, which tends to focus on maximizing the differences between the squared pattern structure coefficients of a factor. Next, a univariate multiple logistic regression was performed based on the extracted factors for males, females, and the total sample. In this step, we adjusted the unit of change to 0.1 for clear interpretation of the OR. All statistical tests were based on *p* < 0.05, and a 95% CI was calculated. All statistical analyses were performed using SAS version 9.4 (SAS Institute, Cary, NC, USA). The overall statistical analysis method is presented in Appendix A.

## 3. Results

### 3.1. Basic Characteristics of the Study Population

We divided the samples into cataract and non-cataract groups and subdivided each group according to sex (Table 1). In the cataract group, there were 1.8 times more females (*n* = 1373) than males (*n* = 764). The prevalence of cataracts was higher in the 70–79-year-olds (53.1%) than in the 60–69-year-olds (22.1%), and females (42.7%) had a higher prevalence of cataracts than males (31.6%). There was a clear difference in the prevalence of cataracts according to sex and age. In both the cataract and non-cataract groups, the food intake and energy among males were higher than that of the females. The distribution of all nutrient variables used in this study is provided in Appendix A. We further checked whether there were statistical differences in nutrient intake between the two groups through a Student’s *t*-test and Cohen’s *d*. The results of the Student’s *t*-test demonstrated significant differences between the cataract and non-cataract groups in the case of most nutrients (*p* < 0.05), except for fat, saturated fatty acids, polyunsaturated fatty acids, omega-3 fatty acid, and omega-6 fatty acid among males, and the omega-3 fatty acid among females. The results of Cohen’s *d* indicated that there was a standardized mean difference of nutrient intake between the cataract and non-cataract groups. For example, the Cohen’s *d* value of vitamin B2 in the female group was 0.36, which indicated that the group’s means differed by 0.36 standard deviations. Thus, a Cohen’s *d* of 0.36 indicated that 64.1% of the cataract group was above the mean of the non-cataract group, 85.7% of the two groups overlapped, and a 60.0% chance existed that a person picked at random from the cataract group would have had a higher score than a person picked at random from the non-cataract group. The results indicated that most variables had a small effect size, and females had a higher Cohen’s *d* value than the males. The results of the Student’s *t*-test and Cohen’s *d* are provided in Appendix A.

### 3.2. Associations between Nutrients and Cataract

We performed a univariate simple logistic regression analysis to identify associations between each nutrient and cataract occurrence. The OR was also investigated with a 95% CI. Among males, water, protein, dietary fiber, calcium, potassium, vitamin B1, and vitamin B3 were statistically significant (*p* < 0.05; Appendix A). Among females, all nutrients except for sodium were statistically significant (Appendix A). We also calculated VIF values to confirm independence between variables. The results indicated that the VIF values of all independent variables did not exceed ten in both males and females (Appendix A).

We performed a univariate multiple logistic regression analysis to identify associations between multiple nutrients and cataract occurrence (Table 2 and Table 3). In this step, we constructed four models according to the adjustment variables. The “Unadjusted model” was not adjusted during the analysis. “Model 1” was adjusted by socio-demographic factors, and “model 2” was adjusted by socio-demographic factors and medical history. “Model 3” was adjusted by socio-demographic factors, medical history, and lifestyle. In males, vitamin B1 was found to be statistically significant in model 2 and model 3, and water was statistically significant in the unadjusted model. It was found that the higher the intake of vitamin B1 (OR_high_mid_ = [0.685, 0.673]) and water (OR_high_mid_ = 0.726), the lower the incidence of cataracts. In females, polyunsaturated fatty acids, vitamin A, and vitamin B2 were statistically significant (*p* < 0.05) in model 1, model 2, and model 3. The lower the intake of polyunsaturated fatty acids (OR_low_mid_ = [1.555, 2.026, 2.063]) and vitamin A (OR_low_mid_ = [1.504, 1.416, 1.430]), the higher the prevalence of cataracts. The higher the intake of vitamin B2 (OR_high_mid_ = [1.364, 1.626, 1.639]), the higher the prevalence of cataracts. Sugar and vitamin B3 were statistically significant (*p* < 0.05) in the unadjusted model. The lower the intake of sugar (OR_low_mid_ = 1.260) and vitamin B3 (OR_low_mid_ = 1.382), the higher the prevalence of cataracts.

### 3.3. Nutrient Factors Associated with Cataracts

We conducted a factor analysis to identify the relationship between the independent variables. First, the rotated factor matrix for 23 nutrients was calculated (Table 4). Then, we grouped the variables into the corresponding factors with the highest factor scores. Twelve nutrient variables—fat, monounsaturated fatty acids, saturated fatty acids, omega-6 fatty acid, polyunsaturated fatty acids protein, cholesterol, vitamin B2, phosphorus, vitamin B3, vitamin B1, and omega-3 fatty acid—demonstrated high factor scores in factor 1. Previous studies confirmed a correlation between fat and vitamin B, and fat intake and cholesterol [59,60]. We named factor 1 the “fatty acids and vitamin B family” in consideration of the commonality of nutrients. Nine nutrient variables that included dietary fiber, potassium, carbohydrates, vitamin B9, water, sugar, iron, calcium, and vitamin C demonstrated high scores in factor 2. Considering the nutrients in factor 2, we named it “polysaccharides and micro-nutrients”. Finally, two variables, vitamin A and carotene, scored the highest in factor 3. Carotene is known as a precursor of vitamin A [61]. Therefore, we named factor 3 the “vitamin A family”.

A univariate multiple logistic regression was performed using the four models to confirm the associations between three factors with cataracts (Table 5). The results indicated that “fatty acids and the vitamin B family” (factor 1) and “polysaccharides and micronutrients” (factor 2) were significantly associated with cataracts in males and females. This result has a meaning similar to that of the intake of polyunsaturated fatty acids and vitamin B2 as an important variable of cataracts among females, as mentioned in the previous section. However, “vitamin A family” (factor 3) was not significantly associated with cataracts.

## 4. Discussion

This study indicates the association between nutrient intake and the incidence of cataracts. It was found that the intake of vitamin B1 was related to cataracts in males, and the intake of polyunsaturated fatty acids, vitamin A, and vitamin B2 was related to cataracts in females. Consequently, our study produced different results for males and females. Previous studies showed differences in food choices and nutritional status between males and females [62,63]. Even in our dataset, males and females had statistically significant differences in all nutrient variables (*p* < 0.001). These differences in nutritional status and dietary habits according to sex led to different results. The results of the analysis were further compared with that of the existing literature. First, a previous study found that a high intake of vitamin B1 (thiamine) lowered the incidence of cataracts [64]. Another case-control study showed that plasma levels of vitamin B1 were relatively low in the cataract patient group compared with the control group [65]. These results are consistent with our findings. Second, previous studies demonstrated that the intake of vitamins and polyunsaturated fatty acids was associated with cataracts [20,21]. They showed that arachidonic acid, one of the polyunsaturated fatty acids, prevents cataract progression. In addition, animal studies have shown that polyunsaturated fatty acids delay the prevalence of cataracts [66,67,68]. This result is similar to our findings that the incidence of cataracts was high when the consumption of polyunsaturated fatty acids was low in females. According to another previous study, the expression of genes involved in inflammatory reactions, including NOS-2 and Cox-2, was enhanced under the control of NF-*k*B [69]. The activation of cPLA2 and Cox-2 increases the expression of PGE2, which induces the activation of EP-1 and leads to a pro-inflammatory cascade in the cataract lens. As a consequence of EP-1 activation, lens proteins aggregate and cause cataracts. Arachidonic acid, a polyunsaturated fatty acid, is a potential pro-inflammatory substrate for cytosolic phospholipase, and the synthesis of PGE2 is catalyzed by Cox-2 from arachidonic acid. As a result, Cox-2 and PGE2 are activated by an increase in arachidonic acid release. Third, previous studies found that retinol, a fat-soluble vitamin in the vitamin A family, had an inverse relationship with cataracts [20,64]. This is consistent with our finding that increased cataract incidence was observed when vitamin A intake was low in females [70]. Finally, in contrast to previous findings, our results demonstrated no relationship between cataracts, sodium, and vitamin C in any of the groups [10,11]. In our study, higher intakes of vitamin B2 were associated with a higher prevalence of cataracts. There is some evidence in the existing literature to support our findings. As a result of the excess consumption of vitamin B2, free vitamin B2 levels are dramatically increased in the blood, tissues, and urine [71,72,73]. This increase causes damage to photoreceptors. Vitamin B2 also produces toxic peroxides and vitamin B2-tryptophan photo adduct, which can impair cells and the liver. In the eyes of research animals, high doses of vitamin B2 were reported to induce damage to retina cells and lead to cataracts [74,75].

Identifying modifiable risk factors related to cataracts is important for improving the health of older Korean adults from the perspective of preventive medicine. This study focused on the intake of nutrients that can be modified through diet control. However, food products may contain beneficial and harmful nutrients simultaneously; for example, beef liver contains vitamin A [76] but also a high content of vitamin B2 [77]. According to our study, vitamin A is associated with cataract prevention, and vitamin B2 is associated with cataract occurrence. Therefore, nutrients associated with cataract prevention and the potential influence of food products containing other nutrients should be considered. Our study has some limitations in that we only found an association between cataracts and nutrient intake and only covered a specific group. First, our findings indicate that the intake of certain nutrients is associated with cataracts, but it does not imply causality [78]. The addition of causality to our findings requires further observations, experimentations, and research. Second, this study does not cover the association between nutrient intake and cataracts in people under the age of 60 years. To better understand the relationship between cataracts and nutrients, it is necessary to analyze them based on various age groups that include middle-aged people. However, per the KNHANES data, the proportion of cataract patients under the age of 60 was very small—244 out of 9559 samples (2.6%)—making it difficult to perform statistical analyses. We believe that this study outlines the nutrient intake patterns associated with cataracts and expect that our work can be combined with other mechanisms of cataract occurrence, leading to further research on cataract prevention.

## 5. Conclusions

This study found statistically significant associations between cataracts and nutrient intake. We increased the reliability by constructing models according to the adjustment for cataract-related variables and presenting the results of each model. Additionally, we investigated whether certain nutrient factors were associated with cataracts by sex. It was found that fat-related nutrients, polysaccharides, and micronutrients are significantly associated with cataracts in males and females. Our findings may help reduce the cost of cataract treatments and surgery by influencing eating habits and diet to prevent cataract occurrence with further experimental studies and observations.

## Figures and Tables

**Figure 1 nutrients-14-04962-f001:**
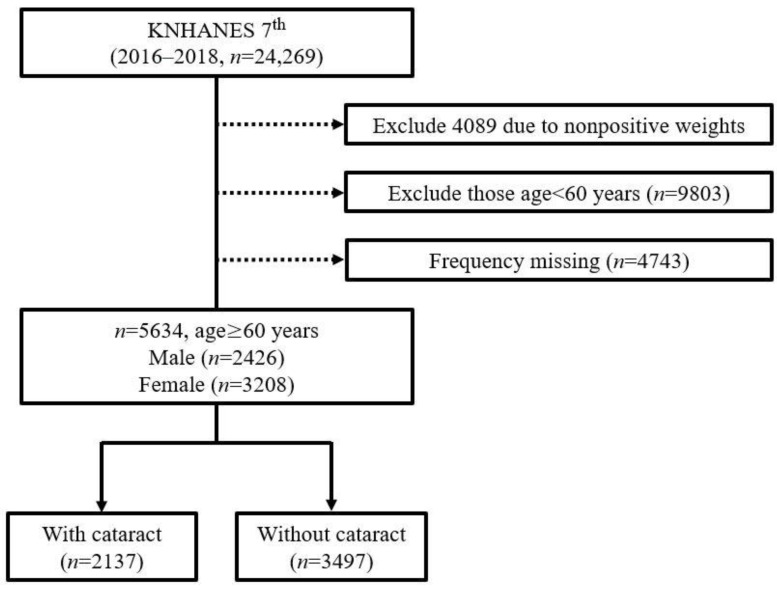
Flow chart for the selection of study participants.

**Table 1 nutrients-14-04962-t001:** Basic characteristics of the study population.

Variables	Unit	Cataract (*n* = 2137)		Non-Cataract (*n* = 3497)	
		Male	Female	Male	Female
Age [*n* (%)]					
60–69		218 (7.9)	391 (14.2)	977 (35.5)	1167 (42.4)
70–79		546 (18.9)	982 (34.1)	685 (23.8)	668 (23.2)
Education [*n* (%)]					
Less than middle school		312 (8.4)	1121 (30.3)	874 (23.6)	1394 (37.7)
Higher than high school		427 (23.2)	204 (11.1)	774 (42.2)	432 (23.5)
National basic livelihood [*n* (%)]					
Non-beneficiaries		74 (13.1)	205 (36.3)	120 (21.3)	165 (29.3)
Beneficiaries		690 (13.7)	1168 (23.0)	1542 (30.4)	1669 (32.9)
Marital status [*n* (%)]					
Single		8 (15.4)	12 (23.1)	19 (36.5)	13 (25.0)
Married		756 (13.5)	1361 (24.4)	1643 (29.4)	1822 (23.7)
Obesity [*n* (%)]					
Absence		357 (14.4)	573 (23.0)	777 (31.3)	779 (31.3)
Presence		225 (13.4)	404 (24.0)	500 (29.8)	552 (32.8)
Hypertension [*n* (%)]					
Absence		293 (12.6)	432 (18.6)	762 (32.8)	837 (36.0)
Presence		469 (14.3)	938 (28.4)	898 (27.3)	995 (29.0)
Diabetes [*n* (%)]					
Absence		325 (11.7)	556 (20.1)	893 (32.2)	995 (36.0)
Presence		439 (15.3)	817 (28.5)	769 (26.8)	840 (29.4)
Asthma [*n* (%)]					
Absence		704 (13.2)	1252 (23.4)	1626 (30.4)	1760 (33.0)
Presence		41 (16.9)	90 (37.2)	36 (14.9)	75 (31.0)
Sinusitis [*n* (%)]					
Absence		701 (13.2)	1262 (23.8)	1592 (30.0)	1756 (33.0)
Presence		44 (16.2)	79 (29.0)	70 (25.7)	79 (29.1)
Allergic rhinitis [*n* (%)]					
Absence		690 (13.4)	1252 (24.1)	1560 (30.0)	1688 (32.5)
Presence		55 (14.0)	89 (22.6)	102 (26.0)	147 (37.4)
Hyperlipidemia [*n* (%)]					
Absence		554 (14.8)	805 (21.5)	1267 (33.9)	1109 (30.0)
Presence		209 (11.0)	568 (30.0)	395 (20.8)	726 (38.2)
Heart failure [*n* (%)]					
Absence		735 (13.2)	1331 (24.0)	1652 (30.0)	1828 (32.8)
Presence		10 (27.0)	10 (27.0)	10 (27.0)	7 (19.0)
Physical activity [*n* (%)]					
Absence		486 (13.0)	975 (26.1)	1033 (27.7)	1239 (33.2)
Presence		246 (13.7)	352 (20.0)	617 (34.0)	581 (32.3)
Smoke [*n* (%)]					
Non-smoker		662 (13.1)	1323 (26.2)	1284 (25.4)	1779 (35.3)
Smoker		137 (23.4)	34 (5.8)	367 (62.6)	48 (8.2)
Heavy drinking [*n* (%)]					
Non-heavy drinking		707 (13.4)	1344 (25.5)	1429 (27.1)	1797 (34.0)
Heavy drinking		50 (15.7)	13 (4.1)	225 (70.5)	31 (9.7)
Nutrient intake (Mean ± SD)					
Food intake	g	1394.1 ± 29.4	1105.0 ± 19.5	1576.9 ± 25.1	1261.8 ± 19.9
Energy	kcal	1892.8 ± 28.7	1421.0 ± 18.9	2026.6 ± 24.0	1558.1 ± 18.9
Water	g	874.3 ± 22.1	722.9 ± 16.1	1003.3 ± 20.1	847.1 ± 16.6
Protein	g	63.7 ± 1.1	46.5 ± 0.7	69.3 ± 1.0	52.3 ± 0.8
Fat	g	31.6 ± 1.0	23.0 ± 0.6	33.9 ± 0.8	26.9 ± 0.6
Saturated fatty acids	g	9.7 ± 0.3	7.0 ± 0.2	10.4 ± 0.2	8.2 ± 0.2
Monounsaturated fatty acids	g	9.4 ± 0.4	6.9 ± 0.2	10.4 ± 0.3	8.2 ± 0.2
Polyunsaturated fatty acids	g	9.4 ± 0.3	7.0 ± 0.2	9.9 ± 0.2	8.0 ± 0.2
Omega-3 fatty acid	g	1.9 ± 0.1	1.4 ± 0.06	1.9 ± 0.07	1.6 ± 0.05
Omega-6 fatty acid	g	7.5 ± 0.3	5.5 ± 0.2	7.9 ± 0.2	6.5 ± 0.2
Cholesterol	mg	162.3 ± 6.1	118.5 ± 4.3	183.3 ± 5.7	138.8 ± 4.5
Carbohydrates	g	315.2 ± 4.8	254.0 ± 3.5	329.5 ± 3.9	272.9 ± 3.4
Dietary fiber	g	27.6 ± 0.7	23.0 ± 0.5	29.4 ± 0.5	25.7 ± 0.4
Sugar	g	51.9 ± 1.6	45.2 ± 1.2	56.7 ± 1.2	52.7 ± 1.1
Calcium	mg	493.8 ± 13.0	400.9 ± 9.8	541.6 ± 10.3	428.9 ± 7.9
Phosphorus	mg	1026.4 ± 17.9	778.8 ± 12.6	1103.6 ± 15.5	860.9 ± 12.2
Iron	mg	12.4 ± 0.3	9.7 ± 0.2	13.3 ± 0.2	10.6 ± 0.2
Sodium	mg	3310.7 ± 76.2	2314.8 ± 46.8	3549.6 ± 63.6	2533.6 ± 50.1
Potassium	mg	2842.4 ± 53.0	2284.2 ± 43.0	3074.1 ± 46.6	2583.1 ± 41.4
Vitamin A	μg	319.2 ± 12.1	261.1 ± 8.3	371.2 ± 17.6	299.7 ± 8.6
Carotene	μg	2809.1 ± 115.2	2248.3 ± 81.2	3151.1 ± 94.8	2630.7 ± 79.5
Vitamin B1	mg	1.3 ± 0.02	1.0 ± 0.02	1.4 ± 0.02	1.1 ± 0.01
Vitamin B2	mg	1.3 ± 0.03	1.0 ± 0.02	1.5 ± 0.03	1.2 ± 0.02
Vitamin B3	mg	11.9 ± 0.3	8.7 ± 0.2	13.2 ± 0.2	10.0 ± 0.2
Vitamin B9	mg	372.7 ± 7.3	271.6 ± 5.5	372.7 ± 6.0	300.4 ± 4.6
Vitamin C	mg	55.7 ± 2.5	53.1 ± 2.3	63.7 ± 2.6	58.7 ± 1.8

**Table 2 nutrients-14-04962-t002:** Results of univariate multiple logistic regression analysis on association between nutrient intake and cataracts for each model, according to adjustment variables in the male group. Each nutrient intake significantly associated with cataracts in univariate simple logistic regression was used as an independent variable.

Nutrition		Unadjusted Model	Model 1 ^a^	Model 2 ^b^	Model 3 ^c^
		Odds Ratio (95% CI)	Odds Ratio (95% CI)	Odds Ratio (95% CI)	Odds Ratio (95% CI)
Water	Low–mid	1.130 (0.839–1.523)	0.966 (0.677–1.379)	1.182 (0.789–1.770)	1.230 (0.800–1.892)
	High–mid	**0.726 (0.548–0.962) ***	0.740 (0.517–1.061)	0.681 (0.435–1.067)	0.669 (0.429–1.045)
Protein	Low–mid	0.930 (0.697–1.239)	0.975 (0.713–1.332)	1.199 (0.830–1.732)	1.148 (0.790–1.669)
	High–mid	0.959 (0.662–1.390)	1.083 (0.724–1.622)	0.979 (0.607–1.581)	0.934 (0.576–1.515)
Monounsaturated fatty acids	Low–mid	1.073 (0.833–1.382)	1.025 (0.778–1.350)	0.920 (0.657–1.289)	0.935 (0.668–1.308)
	High–mid	1.022 (0.663–1.576)	1.146 (0.703–1.868)	1.029 (0.594–1.783)	1.075 (0.622–1.858)
Dietary fiber	Low–mid	0.839 (0.591–1.191)	0.906 (0.614–1.335)	0.854 (0.547–1.335)	0.867 (0.548–1.373)
	High–mid	0.996 (0.759–1.306)	1.023 (0.769–1.360)	1.114 (0.791–1.569)	1.063 (0.755–1.498)
Calcium	Low–mid	1.147 (0.848–1.551)	0.962 (0.696–1.328)	0.889 (0.609–1.300)	0.891 (0.607–1.308)
	High–mid	1.004 (0.749–1.348)	1.009 (0.753–1.353)	1.134 (0.803–1.602)	1.185 (0.838–1.676)
Potassium	Low–mid	0.776 (0.552–1.091)	0.795 (0.552–1.143)	0.954 (0.625–1.459)	0.977 (0.636–1.502)
	High–mid	1.095 (0.819–1.463)	1.087 (0.798–1.479)	1.151 (0.784–1.690)	1.182 (0.803–1.740)
Vitamin B1	Low–mid	1.083 (0.812–1.444)	1.142 (0.832–1.567)	1.115 (0.769–1.618)	1.100 (0.755–1.603)
	High–mid	0.762 (0.561–1.035)	0.763 (0.551–1.057)	**0.685 (0.478–0.981) ***	**0.673 (0.468–0.968) ***
Vitamin B2	Low–mid	0.989 (0.735–1.330)	0.967 (0.698–1.342)	1.152 (0.784–1.693)	1.152 (0.788–1.685)
	High–mid	1.010 (0.742–1.374)	0.998 (0.733–1.358)	1.046 (0.728–1.504)	1.010 (0.697–1.461)
Vitamin B3	Low–mid	1.202 (0.899–1.606)	1.103 (0.814–1.495)	0.873 (0.616–1.237)	0.860 (0.602–1.227)
	High–mid	0.871 (0.636–1.191)	0.914 (0.645–1.294)	0.924 (0.623–1.370)	0.913 (0.613–1.358)

Abbreviations: **Bolded** items indicate a relationship between prevalence of cataracts and nutrient intake. (* *p* < 0.05 ). **^a^** Adjusted for age, education level, national basic livelihood, and marital status. **^b^** Adjusted for all the variables in Model 1, plus obesity, hypertension, diabetes, asthma, sinusitis, allergic rhinitis, hyperlipidemia, and heart failure. **^c^** Adjusted for all the variables in Model 2, plus physical activity, smoking, and drinking.

**Table 3 nutrients-14-04962-t003:** Results of univariate multiple logistic regression analysis on the association between nutrient intake and cataracts for each model according to adjustment variables in the female group. Each nutrient intake significantly associated with cataracts in the univariate simple logistic regression was used as an independent variables.

Nutrition		Unadjusted Model	Model 1 ^a^	Model 2 ^b^	Model 3 ^c^
		Odds Ratio (95% CI)	Odds Ratio (95% CI)	Odds Ratio (95% CI)	Odds Ratio (95% CI)
Water	Low–mid	1.034 (0.798–1.341)	0.928 (0.668–1.290)	0.839 (0.560–1.257)	0.809 (0.537–1.219)
	High–mid	1.009 (0.774–1.316)	1.174 (0.780–1.766)	1.027 (0.628–1.681)	1.032 (0.632–1.685)
Protein	Low–mid	1.049 (0.782–1.408)	1.086 (0.780–1.513)	1.004 (0.688–1.463)	0.992 (0.678–1.453)
	High–mid	0.724 (0.512–1.024)	0.755 (0.527–1.080)	0.707 (0.459–1.088)	0.721 (0.467–1.113)
Fat	Low–mid	1.058 (0.718–1.560)	0.997 (0.660–1.504)	1.090 (0.654–1.817)	1.112 (0.670–1.846)
	High–mid	0.780 (0.448–1.360)	0.652 (0.355–1.196)	0.632 (0.311–1.282)	0.629 (0.310–1.274)
Saturated fatty acids	Low–mid	0.958 (0.726–1.264)	0.902 (0.658–1.237)	0.925 (0.634–1.349)	0.880 (0.607–1.275)
	High–mid	0.970 (0.630–1.493)	1.024 (0.643–1.630)	0.840 (0.463–1.523)	0.857 (0.471–1.557)
Monounsaturated fatty acids	Low–mid	0.802 (0.566–1.137)	0.774 (0.537–1.116)	0.739 (0.490–1.115)	0.745 (0.493–1.126)
	High–mid	0.886 (0.550–1.428)	0.943 (0.554–1.605)	0.991 (0.521–1.886)	1.002 (0.528–1.900)
Polyunsaturated fatty acids	Low–mid	**1.588 (1.113–2.266) ***	**1.555 (1.029–2.349) ***	**2.026 (1.227–3.346) ***	**2.063 (1.256–3.388) ***
	High–mid	1.343 (0.795–2.266)	1.192 (0.668–2.129)	1.047 (0.517–2.120)	1.064 (0.522–2.169)
Omega-3 fatty acid	Low–mid	0.918 (0.714–1.181)	0.967 (0.730–1.280)	0.919 (0.669–1.263)	0.904 (0.656–1.246)
	High–mid	1.099 (0.849–1.422)	1.099 (0.835–1.446)	1.259 (0.909–1.743)	1.244 (0.894–1.732)
Omega-6 fatty acid	Low–mid	0.753 (0.529–1.072)	0.779 (0.538–1.127)	0.677 (0.437–1.049)	0.657 (0.423–1.021)
	High–mid	0.659 (0.401–1.082)	0.794 (0.453–1.390)	0.845 (0.437–1.635)	0.837 (0.432–1.624)
Cholesterol	Low–mid	1.095 (0.873–1.374)	0.944 (0.741–1.203)	1.066 (0.799–1.422)	1.051 (0.786–1.405)
	High–mid	1.135 (0.845–1.525)	1.118 (0.822–1.520)	1.082 (0.740–1.582)	1.053 (0.716–1.550)
Carbohydrates	Low–mid	1.035 (0.765–1.400)	1.163 (0.822–1.646)	1.211 (0.798–1.839)	1.108 (0.726–1.692)
	High–mid	0.929 (0.730–1.182)	0.905 (0.681–1.203)	0.897 (0.644–1.248)	0.927 (0.662–1.296)
Dietary fiber	Low–mid	1.102 (0.815–1.491)	0.990 (0.716–1.369)	0.936 (0.623–1.405)	1.012 (0.669–1.529)
	High–mid	0.901 (0.696–1.168)	0.875 (0.664–1.152)	0.960 (0.694–1.327)	0.956 (0.692–1.323)
Sugar	Low–mid	**1.260 (1.006–1.578) ***	1.221 (0.965–1.546)	1.272 (0.961–1.684)	1.318 (0.995–1.747)
	High–mid	0.941 (0.722–1.227)	1.008 (0.762–1.334)	0.987 (0.711–1.371)	0.990 (0.714–1.373)
Calcium	Low–mid	0.829 (0.636–1.081)	0.800 (0.599–1.068)	0.941 (0.664–1.335)	0.965 (0.681–1.368)
	High–mid	0.931 (0.722–1.199)	0.949 (0.717–1.255)	0.928 (0.670–1.286)	0.938 (0.679–1.296)
Phosphorus	Low–mid	0.824 (0.596–1.138)	0.787 (0.546–1.135)	0.815 (0.527–1.260)	0.822 (0.527–1.283)
	High–mid	1.184 (0.842–1.665)	1.193 (0.829–1.716)	1.201 (0.797–1.809)	1.173 (0.774–1.779)
Iron	Low–mid	0.907 (0.672–1.225)	0.946 (0.678–1.320)	0.898 (0.625–1.290)	0.874 (0.605–1.262)
	High–mid	1.206 (0.931–1.563)	1.291 (0.973–1.712)	1.346 (0.979–1.849)	1.351 (0.977–1.867)
Potassium	Low–mid	1.118 (0.796–1.571)	1.105 (0.756–1.615)	1.153 (0.732–1.816)	1.151 (0.726–1.826)
	High–mid	0.935 (0.717–1.219)	0.983 (0.741–1.304)	0.999 (0.711–1.403)	1.002 (0.713–1.407)
Vitamin A	Low–mid	**1.333 (1.010–1.760) ***	**1.504 (1.109–2.039) ***	**1.416 (1.007–1.992) ***	**1.430 (1.015–2.014) ***
	High–mid	1.173 (0.876–1.571)	1.073 (0.796–1.448)	1.074 (0.741–1.557)	1.047 (0.719–1.524)
Carotene	Low–mid	1.052 (0.833–1.330)	0.957 (0.738–1.240)	1.003 (0.726–1.387)	1.002 (0.723–1.390)
	High–mid	0.929 (0.710–1.217)	1.034 (0.785–1.363)	0.982 (0.718–1.343)	0.986 (0.720–1.350)
Vitamin B1	Low–mid	0.961 (0.719–1.286)	1.098 (0.807–1.496)	0.929 (0.640–1.348)	0.976 (0.672–1.419)
	High–mid	1.164 (0.908–1.491)	1.076 (0.819–1.413)	1.122 (0.809–1.555)	1.126 (0.809–1.568)
Vitamin B2	Low–mid	1.254 (0.935–1.680)	1.144 (0.846–1.547)	1.022 (0.726–1.438)	1.011 (0.718–1.425)
	High–mid	1.199 (0.911–1.580)	**1.364 (1.024–1.815) ***	**1.626 (1.149–2.302) ***	**1.639 (1.160–2.317) ***
Vitamin B3	Low–mid	**1.382 (1.085–1.761) ***	1.234 (0.964–1.579)	1.048 (0.786–1.399)	1.075 (0.805–1.435)
	High–mid	0.898 (0.673–1.196)	1.047 (0.754–1.452)	0.889 (0.606–1.306)	0.850 (0.577–1.252)
Vitamin B9	Low–mid	0.916 (0.666–1.261)	0.806 (0.574–1.130)	0.902 (0.620–1.314)	0.890 (0.609–1.299)
	High–mid	1.009 (0.776–1.313)	1.026 (0.774–1.359)	0.977 (0.698–1.368)	0.983 (0.701–1.378)
Vitamin C	Low–mid	0.996 (0.773–1.284)	1.039 (0.793–1.362)	0.996 (0.716–1.386)	0.995 (0.713–1.387)
	High–mid	1.119 (0.892–1.403)	1.159 (0.910–1.476)	1.078 (0.806–1.441)	1.084 (0.809–1.453)

Abbreviations: **Bolded** items indicate a relationship between prevalence of cataracts and nutrient intake. (* *p* < 0.05 ). **^a^** Adjusted for age, education level, national basic livelihood, and marital status. **^b^** Adjusted for all the variables in Model 1, plus obesity, hypertension, diabetes, asthma, sinusitis, allergic rhinitis, hyperlipidemia, and heart failure. **^c^** Adjusted for all the variables in Model 2, plus physical activity, smoking, and drinking.

**Table 4 nutrients-14-04962-t004:** Results of factor rotation. Factor analysis is a method used to describe unobserved variables for reducing dimensions by analyzing correlations between observed variables. After performing the factor analysis, the results were obtained by rotating the factors to facilitate the interpretation.

Rotated Factor Pattern			
Variables	Factor 1	Factor 2	Factor 3
Fat	**0.95034**	0.15504	0.05081
Monounsaturated fatty acids	**0.91289**	0.09431	0.05992
Saturated fatty acids	**0.86574**	0.09477	0.01638
Omega-6 fatty acid	**0.80332**	0.26433	0.03012
Polyunsaturated fatty acids	**0.79808**	0.28216	0.07921
Protein	**0.74213**	0.46094	0.19707
Cholesterol	**0.68389**	0.14459	0.28426
Vitamin B2	**0.65169**	0.44709	0.33082
Phosphorus	**0.64615**	0.61353	0.25723
Vitamin B3	**0.60773**	0.45240	0.25044
Vitamin B1	**0.52272**	0.46360	0.30187
Omega-3 fatty acid	**0.47650**	0.25455	0.25092
Dietary fiber	0.09142	**0.87826**	0.11848
Potassium	0.35470	**0.80338**	0.26611
Carbohydrates	0.27792	**0.79487**	−0.02306
Vitamin B9	0.20651	**0.75798**	0.34327
Water	0.25510	**0.70240**	0.19341
Sugar	0.27692	**0.62612**	−0.07567
Iron	0.36783	**0.60880**	0.27743
Sodium	0.45774	**0.51476**	0.16612
Calcium	0.33192	**0.51375**	0.34266
Vitamin C	−0.01679	**0.49206**	0.13408
Vitamin A	0.27720	0.13055	**0.84083**
Carotene	0.02572	0.28761	**0.77332**

Abbreviations: **Bolded** items indicate highest factor loadings.

**Table 5 nutrients-14-04962-t005:** Results of univariate multiple logistic regression analysis for each model according to adjustment variables in the male and female groups. Each model used three factors contained by factor rotation as independent variables.

Groups	Variables	Unadjusted Model	Model 1 ^a^	Model 2 ^b^	Model 3 ^c^
		Odds Ratio (95% CI)	Odds Ratio (95% CI)	Odds Ratio (95% CI)	Odds Ratio (95% CI)
Male Group	Factor 1	0.650 (0.302–1.397)	**3.945 (1.560–9.975) ****	**3.367 (1.248–9.088) ***	**3.169 (1.175–8.551) ***
	Factor 2	**0.490 (0.294–0.816) ****	**3.470 (1.740–6.917) ****	**2.783 (1.275–6.074) ***	**2.360 (1.038–5.369) ***
	Factor 3	0.661 (0.379~1.152)	0.750 (0.410–1.371)	0.700 (0.373–1.312)	0.692 (0.372–1.288)
Female Group	Factor 1	**0.278 (0.125–0.618) ****	**4.096 (1.733–9.684) ****	2.395 (0.851–6.735)	2.456 (0.871–6.921)
	Factor 2	**0.371 (0.228–0.604) ****	**6.874 (3.402–13.889) ****	**5.278 (2.400–11.607) ****	**5.074 (1.050–3.519) ****
	Factor 3	0.765 (0.435–1.343)	1.337 (0.766–2.332)	1.347 (0.703–2.580)	1.281 (0.667–2.460)

Abbreviations: **Bolded** items indicate a relationship between cataract prevalence and nutrient intake. (* *p* < 0.05; ** *p* < 0.01). **^a^** Adjusted for age, education level, national basic livelihood, and marital status. **^b^** Adjusted for all the variables in Model 1, plus obesity, hypertension, diabetes, asthma, sinusitis, allergic rhinitis, hyperlipidemia, and heart failure. **^c^** Adjusted for all the variables in Model 2, plus physical activity, smoking, and drinking.

## Data Availability

The data presented in this study are available on request from the corresponding author.

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
