# Peer review of "The Relationship between Nutrient Intake and Cataracts in the Older Adult Population of Korea"

_nutrients, 2022, doi:10.3390/nu14234962_

Round 1
Reviewer 1 Report (New Reviewer)
Very interesting study, very detailed analysis and presentation of the results.
Author Response
Thank you for the reviewer’s comment. We revised the sentences for clarity. In addition, external English proofreading was carried out. We attached the Evidence as below.
Reviewer 2 Report (New Reviewer)
The manuscript entitled „The relationship between nutrient intake and cataract in the older adult population of Korea” presents interesting issue, but some problems should be corrected.
General:
It seems that none of Authors is native English speaker and some sentences are hard to follow and even hard to understand (e.g. “Vitamin B1 was significant in the men’s group” – we do not know what do Authors mean: was vitamin B1 significantly correlated with something in male population – with what?, or was vitamin B1 significantly higher/ lower in male population in comparison with female population? – this is only one example while in the whole manuscript there are various such sentences to be corrected)
Abstract:
Authors should present brief justification of the study.
The description of the studied group should be presented (e.g. what was the sample size, what number of respondents were diagnosed with cataract, etc.)
Authors should clearly indicate what kind of results they present (see above)
Keywords:
Authors should include keywords which are not words from a title to facilitate finding this article
Introduction:
Authors should prepare this section not only to be interesting for Korean readers, but to be interesting for international readers. If Authors prepare their manuscript only for their national readers, they should publish it in some national journal. So, Authors should present here international data from various countries, not only the Korean ones.
Authors should present here detailed information about nutrients and food products influencing (potentially influencing) cataract incidence and course, as well as describe the mechanisms.
Materials and methods:
The questionnaire which was applied should be described (closed-ended or open-ended questions, one-choice or multiple choice questions, etc.).
It should be clearly described how 24-h recall was recalculated into intake of nutrients (the procedure should be described).
It seems that Authors did not verify the normality of distribution and they treated all the variables as normally distributed.
Authors should (1) verify the normality of distribution, (2) for normally distributed data present mean and SD values, but for the other distributions – present median, min and max values, (3) apply adequate statistical tests, that are based on the distribution.
Results:
For normally distributed data Authors should present mean and SD values, but for the other distributions – present median, min and max values.
Authors should apply adequate statistical tests, that are based on the distribution.
The titles of tables should clearly explain what is presented – the titles as “The results of multiple logistic regression using the variables generated through factor analysis. We conducted experiments on the men, and women, respectively” does not explain it.
Discussion:
The discussion can not be based on nutrients only, but it should include the issue of food products containing studied nutrients. As the food products provide multiple nutrients, Authors should address potential influence of food products containing studied nutrients, as it is possible that one product contains at the same time beneficial and harmful nutrient.
The limitations of the study should be presented.
Concussions:
The presented conclusion should be brief and based on the conducted study.
Authors’ contributions:
It seems that contribution of MJL was only minor and he did not participate in preparing manuscript. There is a serious risk of a guest authorship procedure which is forbidden. In such case he should be rather presented in Acknowledgements Section and not be indicated as author of the study.
Round 2
Reviewer 2 Report (New Reviewer)
The manuscript entitled „The relationship between nutrient intake and cataract in the older adult population of Korea” presents interesting issue, but some problems should be corrected.
General:
It seems that none of Authors is native English speaker and some sentences are hard to follow and even hard to understand, even if the manuscript was polished by a professional agency (e.g. “The health interview survey of KNHANES contains four questionnaires related to cataract diagnosis, which are closed-ended questions.” – we do not know what do Authors mean: questionnaires (forms containing a lot of questions), or single questions?)
This manuscript is a resubmission of an earlier submission. The following is a list of the peer review reports and author responses from that submission.
Round 1
Reviewer 1 Report
Have you adjusted for the fact that age and sun exposure can attribute to cataract developement? What about other factors eg. high AND low socioeconomic status that can affect dietary intake
Line 72: Why did you omit participants under 60 years old? i.e. why did you focus on older adults? It can be understood if you omitted children for congenital cataracts, but you will need to justify the 60 years old cutoff. If your hypothesis is nutritional deficiency will result in cataract formation, then age wouldn't matter? Would it be better to stratify by age, including below 60s?
Line 86: Please justify validity of the 24 hour dietary recall method.
Line 109-121: Is there a reason for explanation statistical functions? Are you deviating from convention? If so please emphasis the deviation rather than a paragraph explaining statistical concepts.
Reviewer 2 Report
It is Interesting research, but I think there are some issues that need to be fixed
1. About exclusion criteria, why do you exclude negative weight values? Is there any special reason? If so, I think explanation needed
2. In Figure 1, the number of cataracts and the noncataracts changed.
3. In result, you write that " We divided the samples into cataract and non-cataract groups and subdivided each 171 group according to sex (Table 1). In the cataract group, there were 1.8 times more women 172 (n=1,373) than men (n=764). This result is significant compared to the non-cataract group 173 in which there were 1.1 times more women (n=1,835) than men (n=1,662)." However, looking at Table 1, it can be seen that there is a clear gender difference according to age between the cataract and the non-cataract. Considering the correlation between cataract and age, do you really think this difference is a gender difference?
4. In Table 1, You divided cataract and non-cataract into two age groups: 60-69 and 70-79. And although there is a clear gender difference according to age, there is no explanation of age. Then why did you divide the age into two groups?
